

# Adaptation analysis of two Asteraceae invasive plants in Lhasa, Tibet

Zhefei Zeng[1,2], Ziyi Liang[1], Yan Chen[1], Qi Shu[1], Junru Li[1], Norzin Tso[1], Mengyan Chen[1], Shutong Zhang[1], Xin Tan[1], La Qiong[1,2] and Junwei Wang[1,2]

[1] Key Laboratory of Biodiversity and Environment on the Qinghai-Tibetan Plateau, Ministry of Education, School of Ecology and Environment, Tibet University, Lhasa, China
[2] Yani Observation and Research Station for Wetland Ecosystem of the Tibet (Xizang) Autonomous Region, Tibet University, Nyingchi, China

Corresponding authors
La Qiong, lhagchong@163.com
Junwei Wang, jwyx12240315@126.com

## ABSTRACT

Invasive plants pose a major threat to global ecosystems, especially in ecologically fragile high-altitude regions. Due to its unique geographical and climatic conditions, the Tibetan Plateau is considered highly susceptible to biological invasions. This study investigates the germination capacity and early growth performance of two invasive species, *Bidens pilosa* and *Tagetes minuta*, under the natural climatic conditions of Lhasa, Tibet. We assessed how seed burial depth, geographical provenance, and climatic variables affect their establishment potential. The results showed that *B. pilosa* exhibited the highest germination rate in shallow soil layers, especially for seeds originating from Kunming (KM). However, seeds from the same source that matured in Lhasa following one local growing season (LS), showed a significant reduction in germination capacity, indicating the negative effects of high-altitude stress on maternal seed quality. Seeds from different altitudes displayed varied adaptive performance, with high-altitude provenances showing greater plant height in Lhasa's cold, dry environment. Although *T. minuta* exhibited generally low germination rates across all burial depths, the individuals that successfully emerged demonstrated vigorous early growth, particularly under deeper burial conditions. This suggests that once established, the species may possess strong potential for rapid population expansion and severe invasion. Climate data over the past three years showed that the growing season from May to October in Lhasa—characterized by increased temperature and precipitation—provided a favorable climate window for both species. Our findings suggest that the ongoing warming and humidification trend on the Tibetan Plateau, driven by global climate change, may further facilitate the expansion and establishment of these invasive species. This study provides essential insights for risk assessment and management of invasive plants in Lhasa and other high-altitude regions, emphasizing the importance of long-term monitoring and targeted early-warning strategies in response to shifting environmental conditions.

## INTRODUCTION

Invasive plants represent a significant threat to global ecosystems, posing serious challenges to biodiversity and ecological stability (*Rai & Singh, 2020*; *Shabani et al., 2020*). They

often impact native ecosystems by outcompeting local species, altering habitat structures, and disrupting food chains, leading to the decline or even extinction of native species (*Ayanda, Ajayi & Asuwaju, 2020*; *Bellard, Bernery & Leclerc, 2021*; *Shabani et al., 2020*). For instance, the widespread invasion of *Ageratina adenophora* in southern China and India has led to extensive forest degradation, severely compromising local biodiversity (*Fu et al., 2018*; *Khatri et al., 2023*; *Negi et al.,. 2023a*; *Negi et al., 2024*; *Negi et al., 2025*). *Eichhornia crassipes* poses a serious threat to aquatic ecosystems by rapidly reproducing, covering water surfaces, impeding water flow, and consuming large amounts of oxygen (*Ayanda, Ajayi & Asuwaju, 2020*). Additionally, the invasion of *Ambrosia artemisiifolia* has significantly impacted crop production. Originating from North America, this plant has spread globally, especially in Northeast China, where its competitive nature and suppression of crop growth have led to reduced yields of crops like soybeans and corn, also increasing the cost of weed management in fields (*Knolmajer et al., 2024*; *Qin et al., 2014*). These examples illustrate that invasive plants not only threaten ecosystems but also have severe implications for agricultural production, leading to agricultural yield reduction and substantial increases in the economic costs associated with weed control and management (*Negi et al., 2023b*; *Negi et al., 2024*). In response to the extensive threats posed by invasive plants, researchers globally are actively exploring effective management strategies. However, with the accelerating pace of climate change and globalization, managing invasive plants has become increasingly complex and challenging, necessitating further research and policy support.

China is one of the most biodiverse countries in the world, but due to its geographical and climatic diversity, it has also become a hotspot for alien invasive plants (*Chen et al., 2023*). Currently, 403 invasive plant species have been recorded in China (*Hao & Ma, 2022*), with Asteraceae plants being the most prevalent. These plants demonstrate varied adaptabilities and dispersal capabilities across different ecosystems. The Tibetan Plateau, serving as the "Water Tower of Asia" due to its unique geographical location and climatic features, has become one of the key areas for the study of invasive plants (*Chu et al., 2024*; *Qian & Deng, 2023*). Recent studies indicate that the climate of the Tibetan Plateau is trending towards becoming warmer and more humid (*He et al., 2023*; *Yu et al., 2024*), which may provide new opportunities for the colonization and spread of alien invasive plants. Given the extreme sensitivity of the Tibetan Plateau's ecosystems to environmental changes, the spread of invasive plants could cause irreversible damage to this fragile ecological environment (*Chu et al., 2024*). Therefore, understanding and predicting the distribution of invasive plants on the Tibetan Plateau and their ecological impacts are crucial for protecting the ecological security of the region.

Among numerous studies on invasive plants, species from the Asteraceae family are often the primary focus due to their phenotypic plasticity, strong adaptability and reproductive capabilities (*Hao et al., 2010*; *Martín-Forés et al., 2018*; *Pyšek et al., 2017*; *Yang et al., 2023*; *Khatri et al., 2024*). Asteraceae species typically have rapid growth rates, efficient seed dispersal mechanisms, and good adaptability to various environmental conditions, which give them significant competitive advantages during the invasion process (*Hao et al., 2010*; *Martín-Forés et al., 2018*). Field investigations conducted in the Lhasa River Basin revealed

a notable invasion trend among several Asteraceae species. In particular, *Conyza canadensis*, *Achillea millefolium*, and *Cosmos bipinnatus* have established relatively stable populations in localized areas. Additionally, scattered occurrences of *Bidens pilosa* and *Tagetes minuta* were recorded, primarily along roadsides and agricultural field margins. Notably, both species have shown signs of upward altitudinal expansion within the basin, suggesting a potential for adaptation to higher elevation environments on the Qinghai–Tibet Plateau. *B. pilosa*, native to the Americas, is a globally invasive plant capable of rapidly establishing populations under various environmental conditions. Its seeds have a barbed structure, which facilitates attachment to animals and human clothing, thereby spreading over long distances (*Bartolome, Villaseñor & Yang, 2013*; *Kato-Noguchi & Kurniadie, 2024*). In China, *B. pilosa* has formed severe invasions in southern provinces, posing serious threats to agriculture and ecosystems (*He et al., 2020*). *T. minuta*, also a robust Asteraceae plant native to South America, has formed naturalized populations in various regions of China (*Qi et al., 2022*). Studies have shown its significant allelopathic effects on crops like barley and rice, inhibiting the growth of companion plants and threatening local biodiversity (*Qiu et al., 2020*; *Negi et al., 2023b*).

As the political, economic, and transportation center of the Tibet Autonomous Region, Lhasa experiences frequent tourist activity and active logistics, both of which facilitate the introduction and spread of alien plant species (*Chu et al., 2024*). In a previous floristic survey conducted on the campus of Tibet University in Lhasa, 33 invasive plant species were recorded, though *B. pilosa* and *T. minuta* were not among them (*Wang, Ming & Chen, 2023*). However, subsequent field investigations along the Lhasa River Basin confirmed the presence of these two species in the region. Notably, current species distribution models based on MaxEnt, which rely on climatic variables, do not classify the Lhasa area as highly suitable habitat for either *B. pilosa* or *T. minuta* (*Qi et al., 2022*; *Yang et al., 2023*). This discrepancy may be attributed to a combination of factors, including increased tourism, enhanced connectivity through transportation networks, and ongoing regional climate warming and humidification (*Chu et al., 2024*; *He et al., 2023*; *Yu et al., 2024*). Therefore, a systematic investigation into the ecological adaptability of these species under local environmental conditions is urgently needed to better assess their invasion potential and to inform evidence-based prevention and control strategies for alpine ecosystems.

Seed germination and early seedling development are critical stages in the establishment of plant populations and often determine the success of biological invasions (*Pratap & Sharma, 2010*; *Vibhuti, Bargali & Bargali, 2015*). This study aims to experimentally verify the germination and growth capabilities of *B. pilosa* and *T. minuta* under natural climatic conditions in Lhasa, especially examining their ability to survive the dry, cold, and high UV conditions of winter and to germinate and grow normally in the following year. The study focuses on assessing the effects of different burial depths on germination rates and growth, as well as analyzing the relationship between temperature, rainfall, and the growth of these two invasive plants. Through these experiments, we hope to reveal the adaptability of these invasive plants to the environmental conditions of the Tibetan Plateau and understand the potential impacts of climate change on their spread. The results of this study will provide important information for the management of invasive plants in Lhasa and other similar

ecosystems, a s well as support the development of effective control measures to mitigate or prevent further invasions of these plants into the ecosystems of the Tibetan Plateau.

## MATERIALS AND METHODS

### Seed collection

The *B. pilosa* seeds used in this study were collected from five representative regions across different geographic locations and climatic zones in China, with the aim of assessing the germination capacity and colonization potential of this species in the high-altitude environment of the Qinghai–Tibet Plateau.

In March 2023, we first conducted an outdoor planting experiment at the experimental field of Tibet University (hereafter referred to as the "experimental field"), using seeds previously collected from Kunming, Yunnan Province (designated as KM). This preliminary trial aimed to determine whether *B. pilosa* could complete its full life cycle—germination, growth, flowering, and seed maturation—under the natural climatic conditions of Lhasa. The plants successfully matured and produced seeds by mid-October of the same year. These locally matured seeds were subsequently collected and labeled as LS. This result demonstrated that *B. pilosa* possesses the potential to establish and reproduce in the Lhasa region.

Based on this finding, we further collected seeds from four additional regions to compare the germination performance and early growth responses among different geographic sources. Detailed information on the collection sites of both plants is shown in Fig. 1. Between July and October 2023, naturally matured *B. pilosa* seeds were collected from the following sites: Kunming (KM, the same population as the one used for LS production; elevation ∼1,891 m, subtropical plateau monsoon climate), Mianyang (MY, ∼470 m, subtropical humid climate), Qiandongnan Miao and Dong Autonomous Prefecture (QD, ∼800 m, subtropical monsoon humid climate), and Nyingchi (LZ, ∼3,000 m, warm-humid climate of southeastern Tibet). Among these, MY and QD represent low-elevation regions characterized by warm and humid conditions, whereas KM and LZ correspond to mid- and high-elevation zones located in southwestern China and on the southeastern margin of the Qinghai–Tibet Plateau, respectively, both exhibiting typical monsoonal or warm-humid highland climates. In particular, Nyingchi experiences abundant annual precipitation and has a significantly milder and wetter climate than Lhasa. In contrast, Lhasa is located in the central part of the Qinghai–Tibet Plateau at an elevation of approximately 3,650 m. It has a typical cold semi-arid climate, with cold and dry winters and springs, a short summer rainfall season, intense solar radiation, and high ultraviolet exposure, resulting in harsh environmental conditions overall.

All seeds of *T. minuta* used in this study were collected from Nyingchi (designated as YJ). Nyingchi was selected as the seed source for both species due to its geographic proximity to Lhasa and because both *B. pilosa* and *T. minuta* have exhibited signs of natural establishment in the area. This facilitates evaluation of their potential adaptation and upward expansion into higher elevations. All collected seeds were stored under dry room-temperature conditions in the Environmental Laboratory of Tibet University until
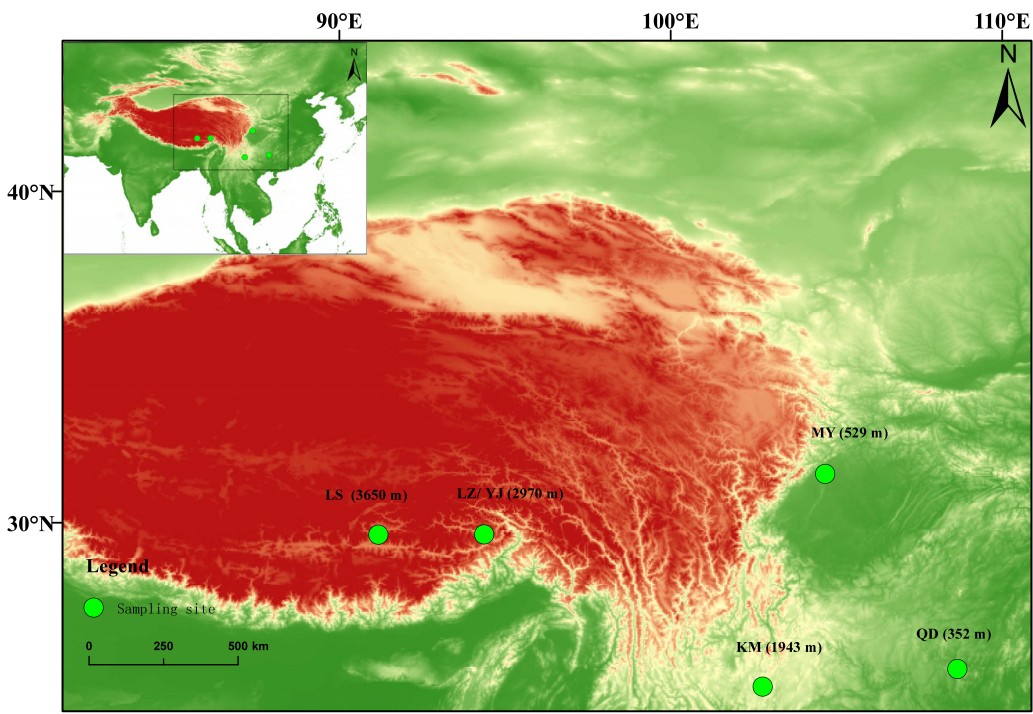

**Figure 1** **Seed collection sites for *B. pilosa* and *T. minuta*.** Green dots indicate the sampling locations, with corresponding elevations shown in parentheses. LS: Lhasa (3,650 m), Xizang Autonomous Region; LZ/YJ: Nyingchi (2,970 m), Xizang Autonomous Region; KM: Kunming (1,943 m), Yunnan Province; MY: Mianyang (529 m), Sichuan Province; QD: Qiandongnan Miao and Dong Autonomous Prefecture (352 m), Guizhou Province. The background is a digital elevation model (DEM), where colors range from green (low elevation) to red (high elevation), illustrating the topographic variation across the region. The map was generated using ArcGIS 10.8.

mid-November 2023 and were subsequently used in an overwintering field experiment conducted in the experimental plots.

## Overwintering field experiment of two invasive plant species

To evaluate the overwintering survival and spring emergence performance of *B. pilosa* and *T. minuta* under natural climatic conditions in Lhasa, an outdoor common garden experiment was established in the experimental field of Tibet University in mid-November 2023. The experimental area is flat, with uniform soil type, ample sunlight, and no shading from buildings or tall vegetation, providing suitable conditions for a field-based ecological study.

## Plot preparation and environmental control

Prior to sowing, the field was plowed and loosened using a small rotary tiller to reduce soil compaction and minimize differences in physical and chemical soil properties among treatment zones. Three parallel sowing strips were established to represent three burial depth treatments: 0 cm (surface sowing), one cm, and three cm.

Visible plant residues on the soil surface were manually removed before sowing, and no herbicides were applied in order to maintain the natural soil conditions. For the

0 cm treatment, shallow furrows were created to provide minimal physical barriers to prevent seed displacement by strong winter winds, without affecting soil moisture or water retention. For the one cm and three cm treatments, shallow trenches were dug to the respective depths, and seeds were sown and then evenly covered with soil to ensure consistency and reproducibility across treatments.

## Experimental design and treatment setup

A completely randomized block design (CRBD) was adopted, with three sowing depth treatments (0 cm, one cm, and three cm). Each treatment included 10 replicates, with each replicate consisting of one seed row containing 50 seeds. Thus, a total of 500 seeds were sown per seed source per depth treatment (50 seeds ×10 replicates), ensuring sufficient statistical power and reproducibility.

Five seed sources of *B. pilosa* (KM, MY, QD, LZ, and LS) and one seed source of *T. minuta* (YJ, from Nyingchi) were included in the experiment. Each seed source was sown in a separate block to avoid mixing, and all plots shared the same soil and climatic conditions to minimize environmental variation. Planting rows were labeled with source identifiers, burial depth, and replicate numbers to ensure accurate tracking during subsequent monitoring.

## Sowing and monitoring

After sowing, no artificial intervention was applied; all seeds overwintered under natural environmental conditions. In the following spring (March to April 2024), systematic surveys were conducted to record seedling emergence rates and plant height across all treatments and replicates. These measurements were used to assess the survival potential and early establishment ability of the two invasive species in the harsh high-altitude environment of Lhasa.

## Data collection and analysis

In the experimental field, the first observation of *B. pilosa* cotyledon emergence occurred on May 15, 2024. However, due to the presence of residual seeds from Cosmos bipinnatus—previously cultivated in the same field—numerous volunteer seedlings emerged in early spring. These seedlings closely resembled those of *B. pilosa* and *T. minuta* during early developmental stages, making it difficult to distinguish species during initial emergence. To ensure the accuracy of species identification, formal germination data collection was delayed until approximately three weeks after sowing. At that time, seedlings had developed their first true leaves, allowing us to reliably distinguish species based on leaf morphology. From this point onward, systematic data collection was conducted. Weekly surveys were performed throughout the growing season to record the number of newly emerged seedlings within each replicate plot. To minimize potential interference and ensure data reliability, all non-experimental seedlings and weedy vegetation were manually removed during the survey period, particularly those occurring within the planting rows. If no new emergence was observed for three consecutive weeks, that date was defined as the germination endpoint for the respective treatment.
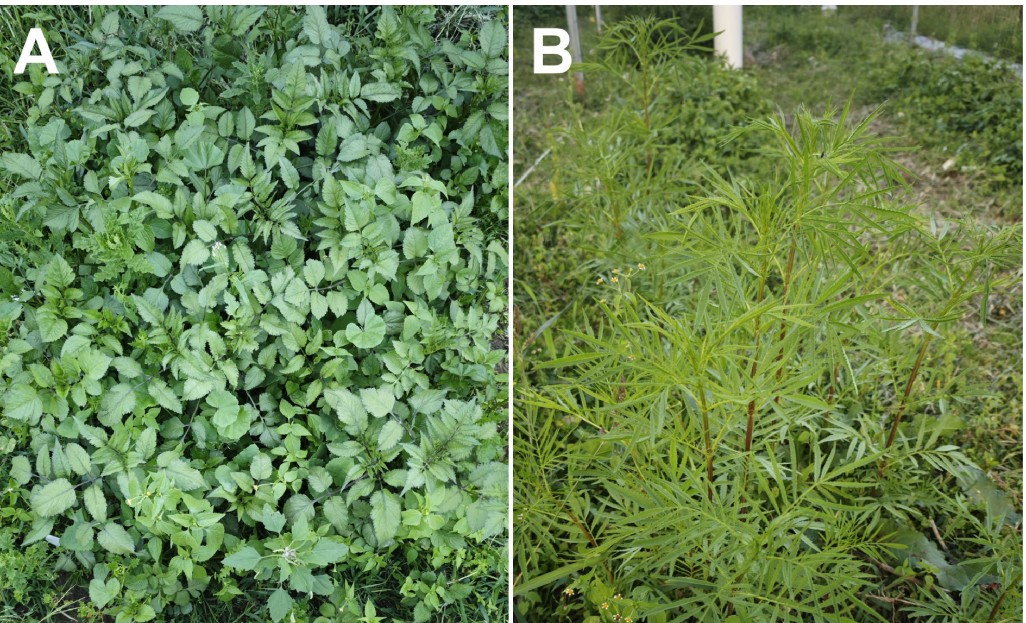

**Figure 2** Photographs of naturally growing *B. pilosa* and *T. minuta* in the experimental field.
(A) *B. pilosa*; (B) *T. minuta*.

To prevent unintended seed dispersal from reproductive individuals, the experiment was terminated on August 15, 2024, when some plants had begun to develop inflorescences and seeds were approaching maturity. On that date, all experimental plants were removed from the field. Prior to removal, plant height measurements were conducted to assess early growth performance. Figure 2 shows the thriving growth of *B. pilosa* and *T. minuta* in the experimental field. For each seed source, 30 vigorous individuals were randomly selected from each burial depth treatment, totaling 90 plants per seed source.

All data processing and statistical analyses were conducted in Python v3.11.4 (The original germination count and plant height data can be found in Table S1). Germination rate was calculated as: (number of emerged seedlings / total number of sown seeds) $\times 100\%$. Plant height data were visualized using boxplots to illustrate distribution patterns and outliers. One-way ANOVA was used to test the effects of burial depth and seed source on both germination rate and plant height. When assumptions of normality and homoscedasticity were met, Tukey's HSD test was applied for post hoc multiple comparisons. Statistical significance was set at $P < 0.05$.

## Microclimate data collection and processing

To explore the relationship between climatic factors and the germination and early growth of *B. pilosa* and *T. minuta* under natural conditions in Lhasa, environmental data were obtained from an SP200 automatic weather station installed adjacent to the experimental field. The station has been in continuous operation since mid-November 2021, providing stable long-term meteorological observations.

For this study, we selected weather records from January 1, 2022, to August 15, 2024, encompassing multiple full annual cycles prior to and during the experiment. This time span offers valuable context for characterizing the climatic background and variability during the experimental period. The weather station recorded data automatically at 15-minute intervals.

To improve data processing efficiency and focus on key climatic variables, we used the Python programming environment (v3.11.4) to clean and process the raw dataset. Daily mean air temperature and daily cumulative precipitation were extracted and used to construct a continuous time series depicting temperature and precipitation trends in the Lhasa area during the study period (see ).

In subsequent analyses, temperature and precipitation data were treated as potential environmental variables and correlated with phenotypic traits such as germination rate and seedling height under different seed source and burial depth treatments. This allowed us to assess the influence of major climatic drivers on the colonization potential of invasive species in high-altitude ecosystems on the Qinghai–Tibet Plateau.

## RESULTS

### Effects of burial depth on seed germination of two invasive Asteraceae species

Figure 3 illustrates the number and germination rates of *B. pilosa* and *T. minuta* under natural climatic conditions in Lhasa, across three burial depth treatments (0 cm, one cm, and three cm) and five seed sources. The results indicate that both seed provenance and burial depth significantly influenced the germination performance of the two species.

For *B. pilosa*, seeds from the original Kunming source (KM) exhibited the highest overall germination, with 301 emerged seedlings and a germination rate of 26.4%. The rates for other sources were as follows: Qiandongnan (QD) at 23.2%, Nyingchi (LZ) at 22.4%, Mianyang (MY) at 17.4%, and Lhasa (LS) at only 11.5%. Notably, the LS seeds were collected from KM plants that had completed one full growing season in Lhasa. The substantially lower germination rate of the LS source may reflect negative effects of the high-altitude environment on seed viability or maternal plant fitness.

Regarding burial depth, all *B. pilosa* seed sources except LZ showed the highest germination rates under surface sowing (0 cm). For instance, the KM source showed significant differences across the three depths, with germination rates of 26.4% (0 cm), 18.2% (one cm), and 15.6% (three cm) ($P < 0.05$). As *B. pilosa* seeds lack dormancy, their germination is likely influenced by external physical factors such as temperature, moisture, and light. Surface sowing can provide enhanced light exposure and better gas exchange, thus facilitating seed germination. These results suggest that *B. pilosa* has high germination potential in the surface soil layer, which may contribute to its successful colonization in novel habitats.

In contrast, *T. minuta* exhibited much lower germination potential overall. The average germination rates across the three burial depths were 4.0% (0 cm), 5.8% (one cm), and 2.2% (three cm), also showing a decreasing trend with increasing depth. Despite the

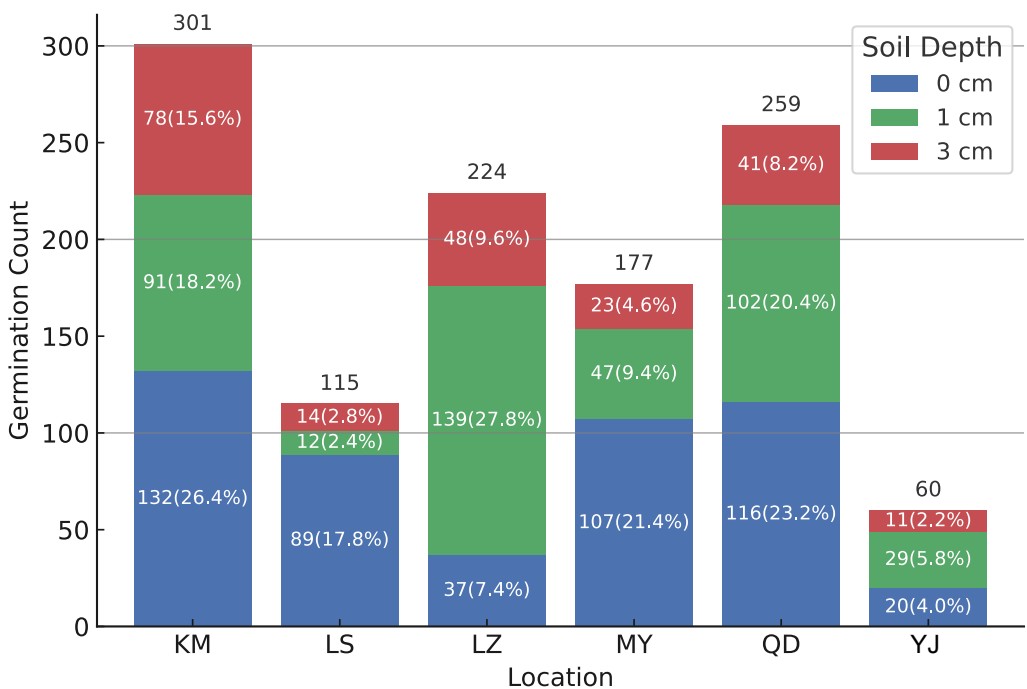

**Figure 3** **Germination performance of *B. pilosa* and *T. minuta* (site YJ) seeds from different sources at various burial depths.** White numbers within each color block indicate the number of germinated seeds and the corresponding germination rates at each burial depth.

low emergence, *T. minuta* is a typical r-selected species with high fecundity and diverse dispersal pathways, including wind, water, and animal-mediated transport (*Wang et al., 2023*). These traits suggest that even with low natural germination rates, the species may still establish and expand rapidly under favorable conditions.

## Effects of burial depth on plant height in two invasive species

Plant height is a key phenotypic indicator of early growth performance and is commonly used in invasion biology to assess a species' colonization potential and competitive advantage (*Holman et al., 2016*). In this study, standardized measurements of plant height were conducted on the final day of the experiment for both *B. pilosa* and *T. minuta* under three burial depth treatments (0 cm, one cm, and three cm) (Fig. 4).

For *B. pilosa*, plant height varied significantly among seed sources. Seeds originating from KM, LS, and LZ—all from high-elevation regions—exhibited greater plant height under surface sowing (0 cm), indicating stronger early growth vigor. In contrast, individuals from the low-elevation sources MY and QD were generally shorter, with minimal height variation across the three burial depths. This pattern suggests that seeds from higher elevations may be better adapted to the cold and arid conditions of the Lhasa Plateau, thus exhibiting more robust growth under similar environmental stresses.

Compared to *B. pilosa*, *T. minuta* exhibited overall taller growth across all burial depths, with consistently higher median values. In particular, some individuals exceeded 70 cm in height under the three cm burial treatment. This marked growth advantage likely reflects

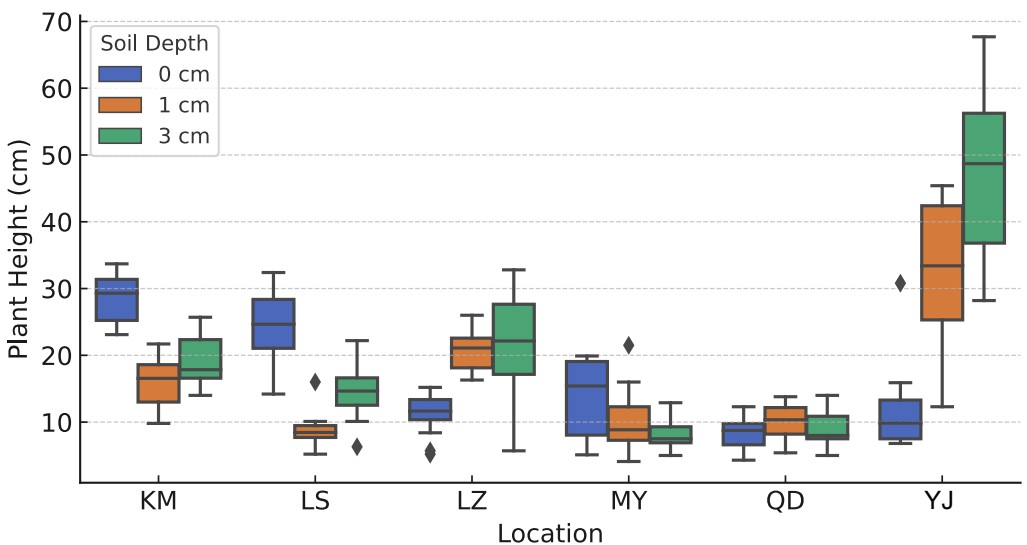

**Figure 4  Variation in plant height of *B. pilosa* and *T. minuta* (site YJ) from different sources across burial depth treatments.**

inherent interspecific differences in growth strategy and morphological development, rather than directly indicating superior adaptation to the high-altitude environment.

## Relationship between germination and climatic factors in two invasive species

The success of plant invasion is often closely linked to local climate conditions and key environmental factors (*Ni et al., 2021*). To evaluate how temperature and precipitation influence the germination of *B. pilosa* and *T. minuta*, we analyzed daily climate data—specifically mean temperature and cumulative precipitation—collected by an automatic SP200 weather station located adjacent to the experimental plots.

Figure 5 presents the daily variation in air temperature and precipitation in Lhasa over the past three years (2022-2024). The data indicate that the primary growing season in Lhasa spans from May to October, during which both temperature and precipitation increase substantially, providing critical conditions for seed germination and seedling establishment (Fig. 5A). In 2024, a sustained rainfall event occurred in early May, with a total of 18 mm of precipitation recorded between May 4 and May 10 (see Table S1), effectively increasing soil moisture levels. This was followed by a rapid rise in daily average temperature, which stabilized above 15 °C by mid-May, creating favorable thermal conditions for seed germination.

Correspondingly, the emergence of seedlings was observed of both species around May 15, suggesting that this period constituted the primary germination window (Fig. 5B). Notably, both species demonstrated considerable tolerance to environmental stress. Despite the harsh winter and spring conditions in Lhasa—characterized by temperatures consistently below 0 °C, minimal precipitation, and intense ultraviolet radiation—experimental observations and climate data confirmed that seeds of both species remained

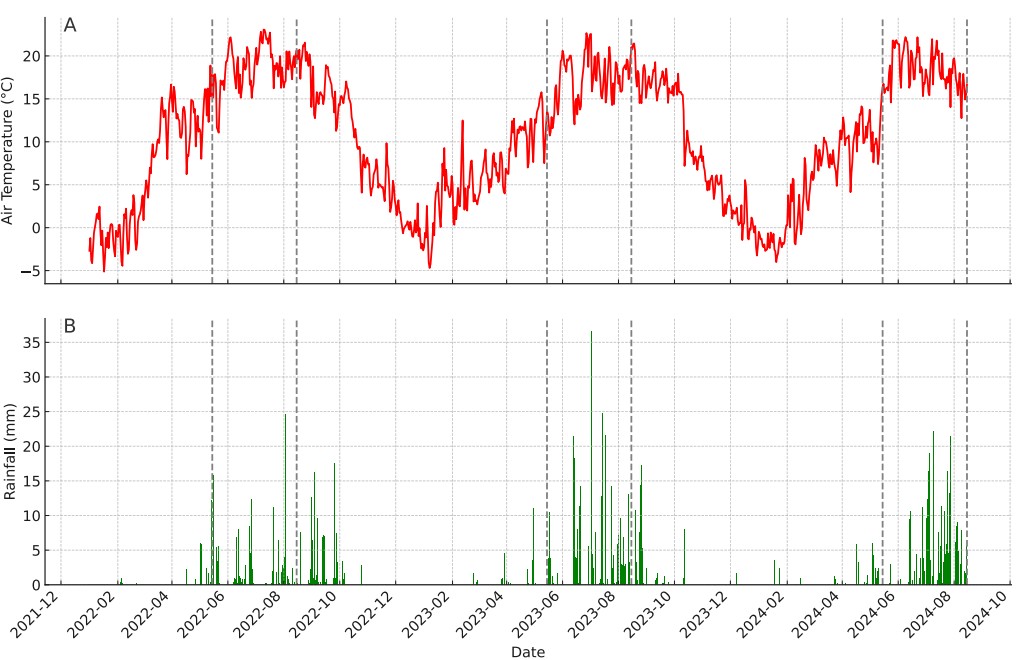

**Figure 5** **Daily air temperature and rainfall trends in Lhasa over the past three years.** (A) Daily air temperature; (B) daily rainfall. The periods corresponding to seed germination and measurement for the two invasive plants are indicated by dashed lines.

viable throughout the winter and were able to germinate rapidly once favorable conditions returned in the spring. These findings suggest that under the ongoing trend of warming and increased precipitation on the Qinghai–Tibet Plateau, the colonization potential of invasive species in this region may continue to rise, thereby increasing their risk of expansion and establishment.

## DISCUSSION

### Effects of seed provenance and maternal environment on germination

This study revealed significant differences in seed germination performance among *B. pilosa* provenances. Seeds originally sourced from Kunming (KM) exhibited the highest germination count, reaching a total of 301 seedlings across the three burial depth treatments, markedly higher than that of the LS provenance (115 seedlings), which originated from KM seeds that had undergone one full growing season and natural maturation under Lhasa conditions. The substantial decline in germination rate of the LS provenance suggests a profound influence of the maternal environment on seed quality. This phenomenon is attributable to maternal effects, where the ecological conditions experienced by the mother plant—such as temperature, radiation, and soil nutrient availability—can significantly affect the developmental quality and vigor of the offspring seeds, subsequently influencing their germination behavior and early growth (*Nguyen et al., 2021*).

In the high-altitude environment of Lhasa, environmental stressors such as low temperatures, high ultraviolet radiation, and reduced oxygen availability likely impair the resource accumulation capacity of mother plants, resulting in reduced seed vigor. In contrast, the mild and humid climate of Kunming is more favorable for maternal development and resource allocation, thus producing seeds with higher viability and germination potential.

Moreover, seeds collected from other external provenances, such as QD, LZ, and MY, also exhibited higher germination rates compared to the LS provenance. This finding suggests that seed populations in the initial stages of invasion may possess greater ecological adaptability. Previous studies have reported that increasing altitude imposes significant constraints on seed germination and seedling survival due to cold temperatures, drought, and intense radiation (*Zait, Konsens & Schwartz, 2020*). Our results further support the view that high-altitude environments exert strong environmental filtering effects on the establishment of invasive plant populations.

## Effects of burial depth on seed germination and plant growth

Burial depth had a significant effect on both germination rate and plant height in *B. pilosa* and *T. minuta*. For *B. pilosa*, seeds from all sources consistently showed the highest germination rates under the 0 cm surface sowing treatment, with a gradual decline observed at one cm and three cm depths. For instance, the germination rates of seeds from the KM provenance were 26.4%, 18.2%, and 15.6% at 0 cm, one cm, and three cm depths, respectively. This trend is consistent with patterns reported in other Asteraceae species. For example, *Artemisia ordosica* exhibited a germination rate of 87.6% at 0.5 cm depth, which dropped sharply to 9.7% at one cm and was nearly zero at three cm (*Zheng et al., 2005*). Similarly, *Xanthium spinosum* showed significantly higher emergence under shallow burial, with maximal rates at 1–3 cm and near-zero germination below five cm (*Tao et al., 2022*). These findings suggest that shallow burial enhances soil warming and diurnal temperature fluctuation near the surface, which promotes germination by accelerating thermal cues, rather than differences in oxygen availability. The surface-sown advantage observed in *B. pilosa* thus reflects a broadly shared trait among Asteraceae species adapted to open or disturbed habitats.

In terms of early seedling growth, *B. pilosa* seeds from mid- to high-elevation provenances (KM, LS, and LZ) exhibited greater plant height under surface sowing compared to those from low-elevation sources (MY and QD), particularly at 0 cm depth. This pattern may reflect a form of pre-adaptation, whereby seeds originating from colder, drier environments possess enhanced vigor under the harsh climatic conditions typical of the Lhasa Plateau.

Although *T. minuta* exhibited generally lower germination rates across treatments, it demonstrated a remarkable growth advantage during the early development stage. In particular, individual plants under the three cm burial treatment reached heights exceeding 70 cm. This rapid growth and structural establishment suggest that once germination occurs under favorable conditions, *T. minuta* could quickly attain a competitive advantage,

highlighting its ecological impact potential despite initial limitations during the germination phase.

## Colonization potential of two invasive species in Lhasa and discrepancies with model predictions

Although the Qinghai–Tibet Plateau is widely regarded as an ''ecological barrier'' to biological invasions due to its high elevation (*Chu et al., 2024*), intense ultraviolet radiation, and arid climate, our findings demonstrate that both *B. pilosa* and *T. minuta* are capable of completing germination and early establishment under natural conditions in Lhasa. Meteorological data revealed a sharp temperature increase and concentrated rainfall beginning in mid-May, which provided a critical ''climatic window'' for seed germination in both species, corroborating the observed synchronized seedling emergence in the field.

However, current ecological niche models (*e.g.*, MaxEnt), which are widely used to predict species' potential distributions, did not identify the Lhasa region as a highly suitable area for either species (*Qi et al., 2022*; *Yang et al., 2023*). This discrepancy may be attributed to limitations in input variables, spatial resolution, and underlying assumptions of the models. Previous studies have shown that invasive plants often overcome original environmental constraints through niche shifts or intraspecific variation (*Petitpierre et al., 2012*). Additionally, microhabitat effects should not be overlooked. Localized moist environments along farmland edges, near buildings, or beside roads in Lhasa may provide more favorable conditions than the macroclimatic averages recorded by meteorological stations. These findings underscore the necessity for cautious interpretation of species distribution model outputs, particularly in dynamic and topographically complex high-altitude ecosystems.

## Invasion risk under plateau warming trends and management implications

Against the backdrop of global climate change, the Qinghai–Tibet Plateau is experiencing a marked trend of warming and increased precipitation, characterized by rising annual mean temperatures, intensified summer rainfall, and an extended growing season (*He et al., 2023*; *Yu et al., 2024*). Our findings demonstrate that seeds of both *B. pilosa* and *T. minuta* remain viable throughout the winter and spring—despite prolonged sub-zero temperatures and minimal precipitation—and are able to germinate rapidly when favorable conditions return the following year. This highlights their strong environmental tolerance and overwintering capacity. In particular, *T. minuta*, as a typical r-selected species, possesses high reproductive potential and diverse dispersal strategies (*Wang et al., 2023*), enabling it to exert significant ecological influence once germination occurs. *B. pilosa* also has the ability to spread over long distances *via* animals or human-mediated vectors. There is already evidence that ecologically similar species have successfully colonized high-altitude regions; for instance, *Ageratina adenophora* has spread widely across the Himalayan Mountains (*Negi et al., 2023a*; *Negi et al., 2023b*). As environmental barriers in the plateau continue to weaken, the risk of plant invasion in Lhasa and across the broader Qinghai–Tibet Plateau is likely to increase.

In summary, under the overarching trend of global warming, the ecological barriers that once limited biological invasions at high altitudes on the Qinghai–Tibet Plateau are gradually diminishing. The potential distribution range of invasive species is likely to shift toward higher elevations and latitudes. As a densely populated and logistically connected city in the plateau region, Lhasa faces heightened risks of exotic plant introductions. Once established, such species may pose long-term threats to native ecosystems. Therefore, we recommend strengthening invasive species monitoring and risk assessment in Lhasa and other key areas across the plateau, with particular attention to high-risk zones that lie outside the range predicted by existing ecological niche models. Future modeling efforts should integrate warming scenarios, species-specific physiological and ecological traits, and microhabitat heterogeneity to improve the accuracy of invasion forecasts and enhance policy responsiveness.

## CONCLUSIONS

This study evaluated the germination ability and early growth of *B. pilos* a and *T. minuta* under natural climatic conditions in the high-altitude environment of Lhasa, Tibet. Our results suggest that seed burial depth, seed provenance, and local climatic factors may significantly influence the early establishment potential of these two invasive species. *B. pilosa* achieved the highest germination rates when seeds were sown at the soil surface, with high-altitude provenances such as KM exhibiting better performance in both germination and early growth compared to locally adapted seeds (LS). This indicates that seed performance may shift between the initial invasion phase and subsequent local adaptation. In contrast, although *T. minuta* exhibited relatively low germination rates, it demonstrated rapid growth and strong vertical development under deeper burial conditions, reflecting its r-selected life-history strategy and potential for ecological competitiveness. Combining our observations with local meteorological data from the past three years, we found that the warm and wet season from May to October provides a climatic window favorable for germination and establishment. These findings are consistent with the emergence of seedlings observed during this period. Under the broader context of global warming, the continued warming and humidification trend across the Tibetan Plateau may further erode environmental barriers, potentially increasing the risk of exotic plant establishment. However, it is important to note that this study only assessed early-stage performance; long-term establishment success and ecological impacts remain to be evaluated through future research.

Overall, our findings provide preliminary evidence that *B. pilosa* and *T. minuta* possess traits conducive to colonization in the Tibetan Plateau, highlighting the need for strengthened surveillance efforts, especially in high-risk areas not currently captured by ecological niche models. Future distribution modeling should integrate climate warming scenarios, species-specific physiological traits, and fine-scale microclimatic data to improve prediction accuracy and support forward-looking management strategies for protecting the region's ecological security.

### Permission for sample collection

*B. pilosa* and *T. minuta* is not listed on the IUCN Red List of Threatened Species, and the collection area is not within a protected zone. Consequently, no specific permissions or licenses were required for sample collection from authorities.

## ACKNOWLEDGEMENTS

We sincerely thank the editors and reviewers for their valuable comments and constructive suggestions, which greatly improved the quality of this manuscript.

### Funding

This work was supported by the National Innovation Training Program for College Students of Tibet University (202410694036), the Tibet Autonomous Region Natural Science Foundation Project (XZ202401ZR0028), the National Natural Science Foundation of China (Grant No. 31760127), and the First-class Discipline Construction Project of Ecology (Zangcaijiaozhi [2023]01). The funders had no role in study design, data collection and analysis, decision to publish, or preparation of the manuscript.

### Grant Disclosures

The following grant information was disclosed by the authors:
National Innovation Training Program for College Students of Tibet University: 202410694036.
Tibet Autonomous Region Natural Science Foundation Project: XZ202401ZR0028.
National Natural Science Foundation of China: 31760127.
First-class Discipline Construction Project of Ecology: Zangcaijiaozhi [2023]01.

### Competing Interests

The authors declare no competing interests.

### Author Contributions

- Zhefei Zeng conceived and designed the experiments, performed the experiments, prepared figures and/or tables, and approved the final draft.
- Ziyi Liang performed the experiments, prepared figures and/or tables, and approved the final draft.
- Yan Chen performed the experiments, prepared figures and/or tables, and approved the final draft.
- Qi Shu performed the experiments, prepared figures and/or tables, and approved the final draft.
- Junru Li performed the experiments, prepared figures and/or tables, and approved the final draft.
- Norzin Tso performed the experiments, prepared figures and/or tables, and approved the final draft.

- Mengyan Chen performed the experiments, prepared figures and/or tables, and approved the final draft.
- Shutong Zhang analyzed the data, prepared figures and/or tables, authored or reviewed drafts of the article, and approved the final draft.
- Xin Tan analyzed the data, prepared figures and/or tables, authored or reviewed drafts of the article, and approved the final draft.
- La Qiong conceived and designed the experiments, authored or reviewed drafts of the article, and approved the final draft.
- Junwei Wang conceived and designed the experiments, performed the experiments, authored or reviewed drafts of the article, and approved the final draft.

## Data Availability

All data used in this study are available in the Supplementary File.

## Supplemental Information

Supplemental information for this article can be found online at http://dx.doi.org/10.7717/peerj.19667#supplemental-information.

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

# PeerJ

**Hao JH, Qiang S, Du KN, Gao YX. 2010.** Wind-dispersed traits of cypselas in ten Asteraceae alien invasive species. *Chinese Journal of Plant Ecology* **34(8)**:957–965 DOI 10.3773/j.issn.1005-264x.2010.08.008.

**He Y, Mo Y, Zheng D, Li Q, Lin W, Yuan G. 2020.** Different sequevars of Ralstonia pseudosolanacearum causing bacterial wilt of Bidens pilosa in China. *Plant Disease* **104(11)**:2768–2773 DOI 10.1094/PDIS-12-19-2738-SC.

**He Z, Zhou T, Chen J, Fu Y, Peng Y, Zhang L, Yao T, Farooq TH, Wu X, Yan W, Wang J. 2023.** Impacts of climate warming and humidification on vegetation activity over the Tibetan Plateau. *Forests* **14(10)**:2055 DOI 10.3390/f14102055.

**Holman FH, Riche AB, Michalski A, Castle M, Wooster MJ, Hawkesford MJ. 2016.** High throughput field phenotyping of wheat plant height and growth rate in field plot trials using UAV based remote sensing. *Remote Sensing* **8(12)**:1031 DOI 10.3390/rs8121031.

**Kato-Noguchi H, Kurniadie D. 2024.** The invasive mechanisms of the noxious alien plant species bidens pilosa. *Plants* **13(3)**:356 DOI 10.3390/plants13030356.

**Khatri K, Negi B, Bargali K, Bargali SS. 2023.** Phenotypic variation in morphology and associated functional traits in *Ageratina adenophora* along an altitudinal gradient in Kumaun Himalaya, India. *Biologia* **78(5)**:1333–1347 DOI 10.1007/s11756-022-01254-w.

**Khatri K, Negi B, Bargali K, Bargali SS. 2024.** Trait plasticity: a key attribute in the invasion success of *Ageratina adenophora* in different forest types of Kumaun Himalaya, India. *Environment, Development and Sustainability* **26(8)**:21281–21302 DOI 10.1007/s10668-023-03529-x.

**Knolmajer B, Jócsák I, Taller J, Keszthelyi S, Kazinczi G. 2024.** Common Ragweed—*Ambrosia artemisiifolia* L.: A Review with Special Regards to the Latest Results in Biology and Ecology. *Agronomy* **14(3)**:497 DOI 10.3390/agronomy14030497.

**Martín-Forés I, Acosta-Gallo B, Castro I, De Miguel JM, Del Pozo A, Casado MA. 2018.** The invasiveness of Hypochaeris glabra (Asteraceae): responses in morphological and reproductive traits for exotic populations. *PLOS ONE* **13(6)**:e0198849 DOI 10.1371/journal.pone.0198849.

**Negi B, Khatri K, Bargali SS, Bargali K. 2023b.** Invasive *Ageratina adenophora* (Asteraceae) in agroecosystems of Kumaun Himalaya, India: a threat to plant diversity and sustainable crop yield. *Sustainability* **15(14)**:10748 DOI 10.3390/su151410748.

**Negi B, Khatri K, Bargali SS, Bargali K. 2024.** Factors determining the invasion pattern of *Ageratina adenophora* Spreng, in Kumaun Himalaya, India. *Environmental and Experimental Botany* **228**:106027 DOI 10.1016/j.envexpbot.2024.106027.

**Negi B, Khatri K, Bargali SS, Bargali K, Fartyal A. 2025.** Phenological behaviour of *Ageratina adenophora* compared with native herb species across varied habitats in the Kumaun Himalaya. *Plant Ecology* **226**:237–249 DOI 10.1007/s11258-024-01487-6.

**Negi B, Khatri K, Bargali SS, Bargali K, Fartyal A, Chaturvedi RK. 2023a.** Impact of invasive *Ageratina adenophora* on relative performance of woody vegetation in different forest ecosystems of Kumaun Himalaya, India. *Journal of Mountain Science* **20(9)**:2557–2579 DOI 10.1007/s11629-022-7862-z.

**Nguyen CD, Chen J, Clark D, Perez H, Huo H. 2021.** Effects of maternal environment on seed germination and seedling vigor of Petunia × hybrida under different abiotic stresses. *Plants* **10**(3):581 DOI 10.3390/plants10030581.

**Ni M, Deane DC, Li S, Wu Y, Sui X, Xu H, Chu C, He F, Fang S. 2021.** Invasion success and impacts depend on different characteristics in non-native plants. *Diversity and Distributions* **27**(7):1194–1207 DOI 10.1111/ddi.13267.

**Petitpierre B, Kueffer C, Broennimann O, Randin C, Daehler C, Guisan A. 2012.** Climatic niche shifts are rare among terrestrial plant invaders. *Science* **335**(6074):1344–1348 DOI 10.1126/science.1215933.

**Pratap V, Sharma YK. 2010.** Impact of osmotic stress on seed germination and seedling growth in black gram (Phaseolus mungo). *Journal of Environmental Biology* **31**(5):721–726.

**Pyšek P, Pergl J, Essl F, Lenzner B, Dawson W, Kreft H, Weigelt P, Winter M, Kartesz J, Nishino M, Antonova LA, Barcelona JF, Cabezas FJ, Cárdenas D, Cárdenas-Toro J, Castaño N, Chacón E, Chatelain C, Dullinger S, Ebel AL, Figueiredo E, Fuentes N, Genovesi P, Groom QJ, Henderson L, Inderjit , Kupriyanov A, Masciadri S, Maurel N, Meerman J, Morozova O, Moser D, Nickrent D, Nowak PM, Pagad S, Patzelt A, Pelser PB, Seebens H, Shu W, Thomas J, Velayos M, Weber E, Wieringa JJ, Baptiste MP, van Kleunen M. 2017.** Naturalized alien flora of the world: species diversity, taxonomic and phylogenetic patterns, geographic distribution and global hotspots of plant invasion. *Preslia* **89**:203–274 DOI 10.23855/preslia.2017.203.

**Qi Y, Xian X, Zhao H, Wang R, Huang H, Zhang Y, Yang M, Liu W. 2022.** Increased invasion risk of *Tagetes minuta* L. in China under climate change: a study of the potential geographical distributions. *Plants* **11**(23):3248 DOI 10.3390/plants11233248.

**Qian H, Deng T. 2023.** Species invasion and phylogenetic relatedness of vascular plants on the Qinghai-Tibet Plateau, the roof of the world. *Plant Diversity* Epub ahead of print 2023 9 January DOI 10.1016/j.pld.2023.01.001.

**Qin Z, DiTommaso A, Wu RS, Huang Hy. 2014.** Potential distribution of two A mbrosia species in C hina under projected climate change. *Weed Research* **54**(5):520–531 DOI 10.1111/wre.12100.

**Qiu X, Luo J, Tu Y, Li H. 2020.** Competitive effect between invasive plant tagetes minuta and native hordeum vulgare. *Chinese Agricultural Science Bulletin* **36**(5):110–114 DOI 10.11924/j.issn.1000-6850.casb18100123.

**Rai PK, Singh JS. 2020.** Invasive alien plant species: their impact on environment, ecosystem services and human health. *Ecological Indicators* **111**:106020 DOI 10.1016/j.ecolind.2019.106020.

**Shabani F, Ahmadi M, Kumar L, Solhjouy-fard S, Tehrany MS, Shabani F, Kalantar B, Esmaeili A. 2020.** Invasive weed species' threats to global biodiversity: future scenarios of changes in the number of invasive species in a changing climate. *Ecological Indicators* **116**:106436 DOI 10.1016/j.ecolind.2020.106436.

**Tao YY, Shang TC, Yan JJ, Hu YX, Zhao Y, Liu Y. 2022.** Effects of sand burial depth on *Xanthium spinosum* seed germination and seedling growth. *BMC Plant Biology* **22**:43 DOI 10.1186/s12870-022-03424-z.

**Vibhuti CS, Bargali K, Bargali SS. 2015.** Seed germination and seedling growth parameters of rice (*Oryza sativa* L.) varieties as affected by salt and water stress. *Indian Journal of Agricultural Sciences* **85(1)**:102–108 DOI 10.56093/ijas.v85i1.46046.

**Wang J, Cao K, Liu J, Zhao Y. 2023.** Biological characteristics and resource utilization of *Tagetes minuta* L. *Journal of Resources and Ecology* **14(3)**:533–541 DOI 10.5814/j.issn.1674-764x.2023.03.009.

**Wang JW, Ming SP, Chen YH. 2023.** Composition and analysis of vascular plant diversity on the Tibetan University campus. *Anhui Agricultural Science* **51(5)**:118–122 DOI 10.3969/j.issn.0517-6611.2023.05.027.

**Yang W, Sun S, Wang N, Fan P, You C, Wang R, Zheng P, Wang H. 2023.** Dynamics of the distribution of invasive alien plants (Asteraceae) in China under climate change. *Science of the Total Environment* **903**:166260 DOI 10.1016/j.scitotenv.2023.166260.

**Yu Y, You Q, Zhang Y, Jin Z, Kang S, Zhai P. 2024.** Integrated warm-wet trends over the Tibetan Plateau in recent decades. *Journal of Hydrology* **639**:131599 DOI 10.1016/j.jhydrol.2024.131599.

**Zait Y, Konsens I, Schwartz A. 2020.** Elucidating the limiting factors for regeneration and successful establishment of the thermophilic tree *Ziziphus spina-christi* under a changing climate. *Scientific Reports* **10**:14335 DOI 10.1038/s41598-020-71276-4.

**Zheng Y, Xie Z, Yu Y, Jiang L, Shimizu H, Rimmington GM. 2005.** Effects of burial in sand and water supply regime on seedling emergence of six species. *Annals of Botany* **95(7)**:1237–1245 DOI 10.1093/aob/mci138.