# Peer review of "Adaptation analysis of two Asteraceae invasive plants in Lhasa, Tibet"

_PeerJ, doi:10.7717/peerj.19667_

## Round 0.1 · original submission · Major Revisions

Reviewer #4 identified a potentially serious problem with the experimental design: It is not clear whether you used replicate plots for each of your seed sources or treatments. If you only used one plot for each seed source, then it becomes impossible to distinguish the effect of the specific plot location versus effect of the seed source on the germination / growth of the plants. The experimental design (including plot layout and spatial arrangement of the treatments and replicates of treatments) needs to be clearly explained. If treatments were not replicated, this work is not publishable.

Specific comments:
Introduction and/or Methods – A clearer explanation is needed to help the reader understand why Lhasa was chosen as the specific study site on the Tibetan Plateau. Does Lhasa differ somehow from your other seed collection site on the Tibetan Plateau (Nyingchi). Why was Nyingchi chosen for seed collection? Does the climate of Nyingchi differ from Lhasa? This should be explained in Methods.

Introduction or Methods – Please explain a bit about how widespread and abundant these two species already are across the Tibetan Plateau.

L 20 “Seeds from local Lhasa sources” – It would be clearer and accurate to say “Seeds from Kumming plants grown in Lhasa…”

L 29 “important scientific evidence” – I suggest replacing with “important information”

L 51-53 “In China, the annual economic loss due to invasive plants is estimated to be in the hundreds of billions (Liu, Li & Zhang, 2012; Senator & Rozenberg, 2017).” – This claim needs to be reworded. There are two issues, 1) the unit for “billions” is not given. Dollars? Yuan? , 2) Even if you are counting in yuan, the numbers given for China in the two references do not add up to hundreds of billions. The first reference counted $3 million actual damage per year (for an insect), while the second reference counted a total cost of around $8 billion for China.

L 103 “important scientific evidence” – I suggest replacing with “important information”


L 117 What are “normal laboratory conditions”? How long were seeds stored?

L 123 What is the “Ecological Garden”? Was soil cleared of plants prior to planting? If so, how was clearing done? Were herbicides used? If not cleared, what existing plants / vegetation occurred in the plots at the time of planting? How did you prevent surface seeds from blowing away over the winter? Are the plots exposed to full sun or do they experience shading from neighboring vegetation or structures? Please provide the size of each plot and the number of replicate plots per provenance. Do the different planting depths occur in different plots of 500 seeds or are the different planting depths located within each plot of 500 seeds? The layout of the experimental plots and replication needs to be clearly explained. If there was no replication, this work is not publishable.

L 134 Approximately how long after germination were you able to identify the species confidently?

L 148 The date of downloading is not important; instead, it is important to tell us the dates (period of time) covered by the weather data.

L 165 Actually, I think LS seeds were not collected in Kunming. Instead LS seeds are progeny of Kunming seeds grown to maturity in Lhasa.

L 172 Unless the soils were waterlogged, there is probably little difference in oxygen between 1 cm and 3 cm. Unless you have oxygen measurements to support the role of oxygen, I don’t think you should attribute differences to oxygen. On the other hand, I could imagine that soil temperature could differ between 1 cm and 3 cm, and that could be relevant. It would have been useful to compare soil temperature at 1 cm vs 3 cm using buried temperature probes.

Figure 3 – Tm is not a location (x-axis label). This needs to be explained in the caption or on the graph.

Figure 4 – Tm is not a location (x-axis label). This needs to be explained in the caption or on the graph.

Figure 5 – On these graphs, please identify the date of planting, date of first germination, and the date when the plant heights were measured (or last germination, end of study)

Figure 5A – I think this is not “Daily temperature changes”, instead it is “Daily average temperature”(?)

L 234 “systematic experimental design” requires replication in order to make valid inferences.

L 243-249 Your findings seem to show evidence of strong maternal effects on germination rates. I think it would be valuable to at least mention the concept of ‘maternal effects’ as there is a large literature on this phenomenon in plants and its importance in ecology.

L 250-252 Your findings comparing LS and KM do not demonstrate a direct effect of UV or moisture in decreasing germination. Both LS and KM were germinated under identical conditions. Instead, the difference seen in germination seems to be due to maternal effects, where mothers who mature their seeds in the LS environment produce less viable or vigorous seeds.

L 263-266 To help provide more evidence for the adaptation hypothesis, it would have been useful to also collect wild LS seeds to compare with the Kumming LS seeds and the LZ seeds.

L 270 I doubt these plants can tolerate low oxygen conditions. I think they will quickly die in anoxic (e.g. waterlogged) soils.

L 277-278 These species’ climactic tolerances have been relatively well-studied from both their native and invasive ranges around the world. Can you please compare your observed climate conditions at LS to other places or studies done on these species? You mentioned in your introduction that modeling suggested that LS is not a suitable climate for these species. Why do you think your results might contradict the species distribution modeling results?

L 297 “study systematically explored” - please remove “systematically” .

L 304 “subsequent adaptation in the local environment” – observed maternal effects are not evidence of adaptation to the local LS environment. None of your findings demonstrated local adaptation to the LS environment - to demonstrate that adaptation has occurred, you would need to identify genetic change after successive generations in the LS environment, and you would also need to demonstrate that those genetic changes led to increased fitness in the LS environment.

L 304-305 This statement seems ok

L 324 “scientific evidence” replace with “information”

One weakness of this experiment is that you did not collect data on flowering and seed production abilities, and I was left wondering whether plants might die before setting seed due to cold/winter conditions. It seems that your Kumming plants did produce seeds in LS in an earlier experiment. It could be appropriate to discuss evidence of successful seed production by these plants on the Tibetan Plateau, either by your field observations or by observations of your own plantings. Such observations would help support your conclusion from the germination plots results, that these species might become invasive on the Tibetan Plateau.

Reviewer 2 ·

Basic reporting

-

Experimental design

-

Validity of the findings

Conclusions are well stated

Additional comments

I have reviewed the manuscript titled "Adaptation Analysis of Two Asteraceae Invasive Plants in Lhasa, Tibet". The study explores the adaptability of Bidens pilosa and Tagetes minuta in the high-altitude environment of Lhasa, Tibet, through germination experiments under natural climatic conditions. The research provides valuable insights into invasive species management in ecologically sensitive regions, contributing to our understanding of plant invasions in high-altitude ecosystems. The paper has potential for publication; however, several issues need to be addressed before it reaches its full scientific rigor.
• The study lacks sufficient details regarding the statistical analysis used to compare germination rates and plant heights. While Python was mentioned, statistical significance and model validation need to be more explicitly addressed.
• The description of the experimental setup does not clarify whether replications were performed to ensure data reliability.
• While the discussion references previous studies on invasive species, it lacks a direct comparison of the findings with existing literature on high-altitude plant invasion.
• The conclusion suggests that Bidens pilosa and Tagetes minuta pose a significant invasion risk, but this claim should be made more cautiously. The study only assesses initial germination and plant height; it does not confirm long-term establishment or ecological impact.

Reviewer 3 ·

Basic reporting

The topic of the paper entitled “Adaptation analysis of two Asteraceae invasive plants in Lhasa, Tibet” is noteworthy and falls within the scope of Peer J. In this manuscript, the authors have evaluated the effect of provenances on the germination of two invasive species along with two environmental gradients using the scientific/technical methods. The research topic and data collected are intrinsically interesting because such types of studies impact the cause-effect relationships between environment, biodiversity as well as society. The research topic and findings are inspiring. Overall, the title clearly reflects the contents. The following suggestions may help to improve the quality of the MS further.

Experimental design

Appropriate

Validity of the findings

Appropriate

Additional comments

The introduction does establish the existing state of knowledge, but needs minor revision as suggested in the reviewed manuscript.
Discussion part needs revision as suggested in the text.
The citations need to be updated.
English needs minor corrections.
I have made more corrections and suggestions directly in the manuscript.
Please revise the paper accordingly.

Annotated reviews are not available for download in order to protect the identity of reviewers who chose to remain anonymous.

·

Basic reporting

Introduction

It is well structured, clearly presenting the threats and damages of invasive plants in terms of economic loss.

The specific case of China and the Tibetan Plateau was presented with respect to the diversity of invasive Plants and their adaptability. The adaptability of the two Asteraceae species studied was highlighted.

The purpose of the study and the usefulness of its results in terms of management of invasive plants, as well development of effective control measures to mitigate or prevent the invasion of the target species, were finally presented.

Materials and methods

Seed Collection

On line 109, they wrote:
”In this study, seeds of B. pilosa were sourced from two locations,” but the following lines (109 – 115) clearly showed that seeds are collected from many locations. This needs to be corrected.

As for Figure 1: Collection distribution points for B. pilosa and T. minuta seeds, it is just a crude extraction of Google Maps with seed point locations.

I recommend them to achieve in GIS, an appropriate and original map where they can locate seed point collections.

Data collection on invasive plants

Figure 2: Naturally growing B. pilosa and T. minuta in the Ecological Garden
It is not just a figure but photos. My suggestion is as follows:
Photos of naturally growing B. pilosa and T. minuta in the Ecological Garden

Mini weather station data collection

No remark!

Results

Germination rates at different seed burial depths

You wrote on lines 164-165:

"These results suggest that seeds from external sources (KM, QD, LZ, MY) had a noticeably higher germination rate than those formed locally from the previous year's invasion (LS)"

That finding is hazardous in the absence of replication of the test and a lack of homogeneity of types of soils across seed sowing points.

You wrote on lines 170-173:
“This trend indicates that B. pilosa seeds germinate better in shallow soil, with deeper burial reducing germination rates, possibly due to restricted oxygen supply and light exposure affecting seed germination.”

Apart from my previous remark that limits the credibility of your results, seeds on the soil surface may benefit from direct sunlight that can contribute to breaking seed dormancy.

Differences in plant height at various burial depths

On lines 188-194, you wrote:

“Specifically, the altitudes for KM, LS, and LZ are 1943 meters, 3650 meters, and 2970 meters, respectively, while MY and QD are only at 529 meters and 352 meters (Fig. 1). The higher altitudes likely confer stronger adaptability on KM, LS, and LZ sources, enabling them to perform better in the high-altitude environment of Lhasa. In terms of plant height at different burial depths, seeds from KM, LS, and MY performed best at the surface level (0 cm), with plant heights significantly higher than at other depths for the same source.”

The factor altitude you referred to could be controlled in a good experimental design. In the absence of such control with adequate replications (at least 2) in the design, your findings can be considered as hazardous, then without consistency, and then reliability. My observations also apply to the variation of plant growth with burial depths.

On lines 200-201, you wrote:

“Compared to B. pilosa, T. minuta plants showed significantly higher plant heights, indicating a faster growth rate than B. pilosa”. This may only be due to the difference in physiology of the two species, hence different responses to environmental factors.

Relationship between two invasive plants, temperature, and precipitation

The periods you referred to in the text are hard to locate in Figure 5. Please, make them more visible.

Discussion

Discussion is structured around seed provenances, altitude, seed burial depths, and climatic factors.

Experimental design

Overwintering planting experiment

The experimental design is not clear enough. They just talk about treatments (sowing depths) and 500 seeds per provenance. Nothing is said about the number of replications and the number of seeds per sowing point. Furthermore, the design should be more characterized. Did they use completely randomized blocks, randomized complete blocks, split plots, etc.?

How did you control the variability of types of soils across the different sowing points?

Validity of the findings

The experiment design of the study is not clear enough. No replication of observations is provided, making the findings quite unreliable

Additional comments

None

---

## Round 0.2 · Minor Revisions

I appreciate careful changes made in response to reviews. I think the manuscript has been greatly improved. I have the following minor comments that should be addressed:

L 27 and L 352 – the manuscript refers to effects of oxygen but oxygen was not measured and the discussion does not cite studies finding a difference in oxygen between 1 cm and 3 cm depth that could affect germination. Please cite such supporting studies (perhaps studies done elsewhere). However, I doubt there was a meaningful difference in oxygen that would affect germination between 1 cm versus 3 cm. In contrast, there was probably a difference in temperature, and temperature is widely recognized to strongly affect germination and growth of plants. Since you did not measure oxygen or temperature, why do you think that oxygen should be singled out in your abstract, but temperature is not mentioned as a possible explanation?

Figure 1 – what is the meaning of the colors on the map? Please explain in the caption.

Figures 3 and 4 – Please explain in the caption that T. minuta is shown as site YJ. e.g. ‘Variation in plant height of B. pilosa and T. minuta (site YJ) from…”

Reviewer 3 ·

Basic reporting

The authors have revised the paper appropriately. I am satisfied with the revised version of the paper and recommended for publication.

Experimental design

Appropriate

Validity of the findings

Okay

Additional comments

Recommended for acceptance

---

## Round 0.3 · accepted · Accept

Thanks for addressing my minor comments in your latest revision.